# Wearable Natural Rubber Latex Gloves with Curcumin for Torn Glove Detection in Clinical Settings

**DOI:** 10.3390/polym14153048

**Published:** 2022-07-28

**Authors:** Norfatirah Muhamad Sarih, Nuur Syuhada Dzulkafly, Simon Maher, Azura A. Rashid

**Affiliations:** 1School of Materials and Mineral Resources Engineering, Universiti Sains Malaysia, Engineering Campus, Nibong Tebal 14300, Penang, Malaysia; fatirahsarih@usm.my (N.M.S.); dn.syuhada@student.usm.my (N.S.D.); 2Department of Electrical Engineering and Electronics, University of Liverpool, Liverpool L69 3GJ, UK; s.maher@liverpool.ac.uk

**Keywords:** natural rubber latex, curcumin, fluorescence-based sensor, torn glove detection, wearable sensor, surgical gloves

## Abstract

Glove tear or perforation is a common occurrence during various activities that require gloves to be worn, posing a significant risk to the wearer and possibly others. This is vitally important in a clinical environment and particularly during surgical procedures. When a glove perforation occurs (and is noticed), the glove must be replaced as soon as possible; however, it is not always noticeable. The present article is focused on the design and development of a novel fluorescence-based sensing mechanism, which is integrated within the glove topology, to help alert the wearer of a perforation in situ. We hypothesized that natural rubber gloves with curcumin infused would yield fluorescence when the glove is damaged, particularly when torn or punctured. The glove design is based on double-dipping between Natural Rubber Latex (NRL) and an inner layer of latex mixed with curcumin, which results in a notable bright yellow-green emission when exposed to UV light. Curcumin (Cur) is a phenolic chemical found primarily in turmeric that fluoresces yellowish-green at 525 nm. The tear region on the glove will glow, indicating the presence of a Cur coating/dipping layer beneath. NRL film is modified by dipping it in a Cur dispersion solution mixed with NRL for the second dipping layer. Using Cur as a filler in NRL also has the distinct advantage of allowing the glove to be made stronger by evenly distributing it throughout the rubber phase. Herein, the optimized design is fully characterized, including physicochemical (fluorescence emission) and mechanical (tensile and tear tests) properties, highlighting the clear potential of this novel and low-cost approach for in situ torn glove detection.

## 1. Introduction

Surgical gloves act as a physical barrier between the surgeon’s hands and the patient, which helps to prevent contamination and disease transmission. Gloves can be damaged during a surgical procedure, potentially harming both the surgeon and the patient. For the most part, glove perforation goes unnoticed by the wearer [1]. Being able to detect glove perforation would significantly aid the surgeon in determining when glove replacement is necessary. Perforations can significantly increase the risk of surgical site infection and expose the surgeon to blood-borne illnesses, such as HIV, hepatitis C, and hepatitis B [2,3].

Glove perforations occur commonly during surgical operations. Orthopaedic surgical procedures have the highest glove perforation rate compared to other types of surgery [4,5,6]. In orthopaedic surgery, the use of power equipment, handling sharp bones, and working in deep holes all contribute to the risk of glove perforation [4,6]. Various reports suggest that the causes of glove perforation include general wear after a given time period of use, hand dominance and, as already noted, the impact of certain surgical procedures [5].

One way to combat glove perforation is through ‘double glove’ use, whereby the wearer will put on at least two pairs of gloves. A glove perforation rate study revealed that the total perforation rate between single and double glove sets is 15.2% and 14.4%, respectively [7]. It was shown that double gloving yielded 141 perforations from 512 glove sets, whereby the number of inner gloves perforated as a consequence of puncture from the outer to the inner gloves was 1.17%. Thus, even if the outer gloves were perforated, double gloving provided 98.83% protection. Another study reported that using double gloves for surgery resulted in an inner glove perforation rate of 3.7% [8]. The double gloving approach to manage glove perforation has been shown to be effective and provide greater than 90% protection for both the patient and the surgeon. However, given the potential life-threatening consequences of torn gloves, this rate is still far from ideal and wearing two pairs of gloves can compromise tactile sensitivity, manual dexterity and two-point discrimination [9]. Avoiding glove perforation is perhaps impossible; consequently, it becomes necessary to provide a fail-safe so that when tears do inevitably occur they can be quickly identified and remedied. According to Ashton et al., identifying certain surgical processes or scenarios when glove contamination is high may assist the surgeon in determining whether a glove change is required [10]. Needless to say, surgeons work under extreme pressure and stressful conditions; thus, checking for glove defects whilst undergoing a surgical procedure is a concern that we envision can be removed. Rather than doctors being aware of certain surgical scenarios that may result in damaged gloves, we suggest that wearing an integrated glove-damage detection device can provide rapid feedback in the event of a tear, allowing swift remediation, significantly reducing the risk of infection. Hence, in this work, we have sought to design and develop a novel glove topology that can provide the wearer with instant visual feedback when perforation occurs.

Curcumin is a polyphenolic compound mainly found in turmeric (*Curcuma longa* L.). It has received increasing interest from numerous studies owing to its health benefits such as anti-inflammatory, anti-diabetic, anti-microbial and anti-oxidant properties [3], as well as being relatively safe for human use [11]. Curcumin has good optical properties because of the symmetric structure of delocalized π electrons. Two benzene rings at the terminal of the curcumin molecule are joined by a seven-carbon heptadiene chain comprising a ß-unsaturated ß-diketone structure that shows keto-enol tautomerism (Figure 1), resulting in an extensive large delocalized-conjugated system and rigid planar structure [11]. Because of its optical properties as a fluorescent polyphenol, curcumin can be employed as a sensing material for the detection of chemicals [12]. Curcumin-based fluorescent probes can solve the lack of organic fluorescent dyes, for example, low quantum yield, and poor photostability [13,14]. Furthermore, since curcumin possesses multifunctional based fluorescence probes with high sensitivity, selectivity, and stability, much research has been published on chemosensor [15,16,17,18,19,20,21,22,23,24] and biosensor [25,26,27,28,29,30,31,32] applications of curcumin [12]. Thus, a strong body of foundational knowledge relating to the use of this compound is readily available.

In previous work, we have shown how NRL can be modified to improve its antimicrobial properties for potential use as a stethoscope diaphragm cover [33]. In this research, we focus on the design and development of a novel torn-glove fluorescence-based sensor by filling curcumin in NRL compounding. The fluorescence of the curcumin coating/dipping layer beneath NRL helps to indicate the appearance of tearing or perforation under UV light. UV lights are commonly used in hospitals and especially in the operating theatre for surface and air disinfection purposes. Jerry et al. reported that using UV lights in the operating theatre during total joint replacement surgery appears to be an effective technique to reduce infection risk when necessary safety precautions are performed [34]. UV light in the operating room is equally effective as laminar airflow in reducing the number of bacteria and infection rates in the operating room by destroying bacteria [35]. Conceivably, the production of a torn glove fluorescence-based sensor will help to detect any perforation damage to the glove under the illumination of UV light.

## 2. Materials and Methods

### 2.1. Materials and Instrumentation

Curcumin (Cur) dispersion was extracted from turmeric by ethanol extraction and centrifuged for 15 min at 6000 rpm. Ethanol (98–100%) was acquired from Merck. The compounded NRL was purchased from Getahindus Sdn. Bhd., Malaysia. The total solid content (TSC) of compounded NRL is 60.8% and was diluted to 44% by adding distilled water. The latex compounded total solid content (TSC, %) was determined according to ISO 2004:2005 (E) Fifth Edition 1 September 2010.

### 2.2. Preparation of NRL Filled with Curcumin (NRL-Cur Mixture)

In total, 6.7% (TSC) of curcumin dispersion was mixed with NRL at 0.5 phr. Distilled water was added to the NRL-Cur compounding for dilution to TSC of 26%. The mixture was then stirred for 2 h and left for latex maturation at room temperature for 24 h before being used.

### 2.3. Preparation of Double Coating Films (**LC**)

The double coating films, which are NRL as the first dipping and NRL-Cur mixture as the second dipping (**LC**), were prepared firstly by the coagulant dipping process. At first, a cleaned and dried porcelain former was dipped into a slurry coagulant of 10% calcium nitrate solution for 10 s and dried in an oven at 70 °C for 10 min. Then, the former was dipped into the NRL compound based on the dwell time mentioned in Table 1, followed by drying in the oven at 100 °C for 15 min. The dried NRL films were taken out from the oven for ~5 min until the film temperature reached 55–60 °C. Then, the former continues to be dipped in NRL-Cur mixture for a few seconds (as dwell time stated in Table 1) and dried in the oven for 15 min at 100 °C before stripping out from the former using calcium carbonate powder. The thickness of each test piece was measured using a thickness gauge (Mitutoyo Corp., Kawasaki, Japan, Model Mitutoyo 7301) and the average thickness was taken. The thickness of film preparation needs to be controlled in the range of 0.20–0.26 mm by varying the dipping dwell time.

### 2.4. Post-Processing of **LC** Film

Three optimum samples of **LC** films were chosen for post-processing. In the post-processing stage, the impact of various leaching situations on mechanical properties and fluorescence of **LC** films were investigated. The leaching procedure included wet-gel leaching, dry-gel leaching, and a combination of wet and dry gel leaching. This process is essential for producing NRL gloves because it removes soluble protein and excess chemicals [36]. For wet-gel leaching condition, after second dipping of NRL-Cur, the dipping plates were allowed to cool for 10 min before being wet-gel leached in distilled water at 70 °C for 1 min. Then, **LC** films were vulcanized and stripped. For dry-gel leaching, the **LC** films were stripped from the plates after vulcanization, and then they were soaked in distilled water for 1 h at room temperature. While, for a combination of wet and dry leaching, before vulcanization, the **LC** films were leached in distilled water at 70 °C for 1 min. Then, after stripping, the **LC** films were re-leached in distilled water for 1 h at room temperature. Figure 2 illustrates the leaching process for the optimised **LC** film. Leaching is a critical technique in the production of medical gloves because it is the most practical way to reduce undesired elements in rubber gloves while maintaining a clean product. Effective elimination of extractable proteins (which cause Type 1 allergy) and residual chemicals (which cause Type IV skin irritation) requires a mix of wet gel and dry leaching methods [37]. The mechanical properties and fluorescence of the **LC** films were determined for each leaching condition.

### 2.5. Characterization of Mechanical Properties

Dumbbell shaped test pieces were cut from the LC films. The tensile test was done following the ASTM D416 standard using a computerized tensile tester machine (Instron Corp., Norwood, MA, USA, Model Instron 5564). The crosshead speed of the Instron machine was set at 500 mm/min until the test pieces failed. The values of tensile properties were the average of 5 measurements. A tear test was conducted according to the ASTM D624 standard. The **LC** films were cut into angle test specimens in the dipping direction. The tear test was carried out using an Instron machine with a crosshead speed of 500 mm/min ± 50 mm/min until the test pieces failed. The tear strength was obtained, and the average of 5 readings was recorded.

### 2.6. Fluorescence Spectroscopy of **LC** Films

Fluorescence analysis of the **LC** films was measured by using a Fluorescence Spectrometer (PerkinElmer Inc., Waltham, MA, USA, Model PerkinElmer LS 55 Fluorescence Spectrometer). The excitation wavelength was set at 380 nm, and the fluorescence emission spectrum was scanned from 400 to 600 nm with a scan rate of 50 nm s^−1^. The **LC** films were cut into a circle (~2 cm diameter) and placed in the solid compartment of the fluorescence spectrometer. Four v-shaped lines were cut (~1.40 cm to each line) into the films so as to simulate a puncture or tear, as illustrated in Figure 3. The cuts in the NRL film are clearly visible when examined under UV light (380 nm) and this is also evident in the corresponding fluorescence emission spectra results (see Section 3.2 and Appendix A).

### 2.7. Characterization of **LC** Films

Fourier Transformed Infrared Spectroscopy (FTIR) spectra of the samples were recorded using a FTIR Spectrometer (Bruker Optics, MA, USA, Model Alpha) in transmission mode with the wavenumber ranging from 4000 to 500 cm^−1^ for sample functional groups characterization.

### 2.8. Thermogravimetric Analysis

Thermogravimetric analysis (TGA) was observed using a TGA analyzer (Perkin Elmer, MA, USA, Model Pyris 6) with a temperature range from 30 to 600 °C at a heating rate of 20 °C/min under a nitrogen atmosphere at a flow rate of 10 mL/min. The weight loss was calculated from the initial and final weight of the thermograms obtained as follows (Equation (1)):(1)$$\;\mathrm{Weight}\;\mathrm{loss}\;\left(\%\right)=\frac{\left(\mathrm{Initial}\;\mathrm{weight}-\mathrm{Final}\;\mathrm{weight}\right)}{\mathrm{Initial}\;\mathrm{weight}}\times 100$$

## 3. Results and Discussions

### 3.1. Mechanical Properties of the **LC** Films

Table 2 shows **LC** films’ tensile and tear strengths with a different dwell time of NRL and **LC** mixture dipping. All the **LC** films showed better tensile and tear strength than the control sample of NRL alone. The tensile strength of the **LC** films for “5_5” dwell time of curcumin NRL-Cur mixture dipping (**LC1**, **LC3**, **LC5**, **LC7**) showed better strength than 5 s dwell time films (**LC2**, **LC4**, **LC6**, **LC8**). In contrast, the tear strength of **LC2**, **LC4**, **LC6**, and **LC8** (5 s dwell time) films was better than that of **LC1**, **LC3**, **LC5,** and **LC7** films (“5_5” dwell time). It showed that increasing the dwell time of curcumin can provide better tensile strength but is still good in tear strength when compared to quick dipping (5 s, in and out) with the NRL-Cur mixture. This is due to the increase in film thickness caused by the accumulation of more latex on the former. As film formation time increases, more latex particles are able to settle and form continuous films, as evidenced by the increased crosslinking observed in cured NRL films [38]. However, the differences in tensile and tear strength for each **LC** film based on the “5_5” and “5” dwell times of **LC** mixture dipping are less than 1.75 MPa and 5.23 N/mm, respectively, which is a slight difference when compared to each similar dwell time of NRL dipping. The minimum allowable tensile strength values prescribed by ASTM standards range from 14 to 24 MPa, which depends on the glove type. Thus, the tensile strength data (Table 2) showed that the values are still within the specified range by ASTM standards.

### 3.2. Fluorescence Effect of the Torn/Cut **LC** Films Based on Dwell Time of Dipping

Each of the films was fluorescence examined based on solid sample analysis. Based on the maximum fluorescence intensity in Figure 4 and the fluorescence spectrum of each film (Appendix A), it showed that the films’ fluorescence intensity increased after tearing. Besides, dipping patterns’ dwell time is also correlated to the fluorescence intensity of the torn/cut **LC** films. The longer the dwell duration of the **LC** mixture dipping, the higher the intensity of the fluorescence. For example, for the “5_5” dwell time **LC** mixture dipping for the torn sample, **LC-T1** has higher fluorescence intensity than the “5” dwell time for film **LC-T2**, as illustrated in Figure 4. In this work, the dwell time for **LC** mixture dipping was confined to less than the dwell time of the first dipping of NRL. It is to conceal the appearance of the inner underneath film colour from the outside layer of NRL film since curcumin is obviously yellow with a strong yellowish-green fluorescence emission. Moreover, when compared based on the dwell time of NRL latex dipping, most of the **LC** films do not show much difference, especially with the similar dwell time of **LC** mixture dipping. As shown in Figure 4, torn films (T) of **LC1**, **LC5**, and **LC9** had nearly identical maximum fluorescence intensities, similar to **LC2**, **LC6**, and **LC10** films. However, torn films of **LC3**, **LC4**, **LC7** and **LC8** exhibited the largest fluorescence intensity. Based on this observation, the 5 secs dwell time of NRL dipping produced an increased fluorescence emission when torn than other dwell times. We suggest that this is due to the thicker NRL film layer, which yields an increased fluorescence effect when torn; other works have indicated a positive correlation between fluorescence intensity and film thickness, which we suspect is possibly the case here [39,40]. Even though “15_5_15” dwell time (**LC7** and **LC8**) showed the highest emission intensity when torn, the thickness (0.25–0.26 mm) of the films were quite thick for glove production. However, it is still less than the maximum thickness of a surgical glove as specified by international standards [41].

### 3.3. Effect of Leaching Process on Mechanical Properties of the **LC** Films

Post-processing of the **LC** films is necessary to identify the effect of leaching. Therefore, the study was carried out to explore the optimum films (**LC1**, **LC3** and **LC7)** on the mechanical properties of **LC** films and their fluorescence after the films were torn/cut. The films were chosen because they gave better results than other films when compared in mechanical tests, fluorescence intensity and the thickness of the film (which is within the recommended range for surgical glove thickness). The tensile and tear strengths of the **LC** films at different leaching conditions are shown in Table 3. Typically, leaching could increase the tensile strength of NR latex due to the elimination of water-soluble non-rubber substances, primarily proteins, and increase the efficient coalescence of rubber particles. As a result, leaching the proteins caused improved interaction between rubber molecules from nearby particles, resulting in a more coherent film [42]. Moreover, the excess calcium nitrate and water-soluble non-rubbers, such as additional compounding components, are also removed, which improves physical properties [43]. As shown in Table 3, the leaching **LC** films displayed greater tensile strength than the unleached NR latex film. The tear strength of the filled NR latex films increased after the leaching procedure. A combination of wet and dry gel leaching provided the best mechanical properties in tensile and tear strengths.

### 3.4. Effect of Leaching on Fluorescence Analysis of **LC** Films

The fluorescence emission intensity at 510 nm for **LC1**, **LC3** and **LC7** films after leaching are shown in Figure 5. It is demonstrated that wet leaching samples gave the highest fluorescence intensity, exhibiting a significant difference in intensity between the control and torn films. Moreover, each leaching sample gave higher intensity after being torn/cut than the control. Although, the dry leaching films show small gap difference in fluorescence intensity due to the leaching condition. The residue of curcumin dye from the curcumin-latex layer during dry leaching may affect the layer of NRL because the whole films were leaching after stripping; compared to wet leaching, only the curcumin-latex layer was leached in the water. After comparing leaching and without leaching, the fluorescence intensity of leaching films is higher than without leaching. This comparison can also be observed visually in the images of the films under UV light in Table 4.

### 3.5. FTIR Study

Figure 6 illustrates the FTIR spectrum of NRL film, **LC** films, and curcumin. The characteristic peaks of cis-1,4-polyisoprene, at 2960 cm^−1^, 2925 cm^−1^, 2850 cm^−1^ (C–H stretching), 1661 cm^−1^ (C=C stretching), 1442 cm^−1^ (C–H bending of –CH_2_–), 1374 cm^−1^ (C–H bending of –CH_3_) and 841 cm^−1^ (C–H bending of C=C–H), characterize the NRL characteristics bands [44]. Curcumin showed its characteristic peaks at 3500–3000 (phenolic O–H stretching vibration), 1626 cm^−1^ (aromatic moiety C=C stretching), 1585 cm^−1^ (benzene ring stretching vibrations), 1508 cm^−1^ (C=O and C=C vibrations), 1418 cm^−1^ (C–H bending vibrations), 1260 cm^−1^ (aromatic C–O stretching vibrations), and 1016 cm^−1^ (C–O–C stretching vibrations) [45]. Meanwhile, **LC** exhibited peaks at 3500–3000 (phenolic O–H stretching vibration), 1641 cm^−1^ (aromatic moiety C=C stretching), 1578 cm^−1^ (benzene ring stretching vibrations) and 1535 cm^−1^ (C=O and C=C vibrations), while enol C–O peak was obtained at 1249 cm^−1^, and C–O–C peak at 1012 cm^−1^, which are the characteristic peaks of curcumin, indicating the existence of curcumin in the NRL films.

### 3.6. Thermal Properties by Thermogravimetric Analyzer (TGA)

Thermogravimetric analysis is a method for determining the mass change of a sample over time or at a specific temperature in a controlled environment. The measurement is generally used to ascertain the material’s thermal and oxidative stabilities and its compositional properties. The weight losses due to heat degradation of NRL and **LC** films are depicted in Figure 7. The weight losses were calculated as a function of temperature. The first stage of deterioration of the NRL began at around 58 °C, whereas **LC** at 220 °C since the absorbed water evaporated and ended at 450 °C owing to the thermal decomposition of the rubber matrix. The mass loss peak found between 450 and 580 °C was attributed to the rubber’s heat breakdown of carbonaceous residues [46]. The temperature NRL and **LC** films at 5%, 50% and 75% weight loss, respectively, are shown in Table 5. As can be observed, the temperature of **LC** film steadily increased at 5% and 50% weight loss than in NRL. However, then, the temperature decreased after 75% weight loss of **LC** film and yielded 100% total weight loss at 538 °C. It showed that the thermal stability has reduced due to the presence of curcumin, a higher unsaturated compound with more double bonds than NRL, which caused slightly lower oxidative stability.

## 4. Conclusions

In this research article, we presented a novel wearable glove design that combines latex and curcumin natural dyes for detecting torn or punctured gloves. We have demonstrated that the latex-curcumin (**LC**) layer can provide a clear optical signal to indicate that the material has torn. Further, our glove is made from low-cost materials, is remarkably simple to fabricate using curcumin extracted from turmeric and is safe to wear. The **LC** films’ mechanical properties were comparable to the control sample’s tensile strength but with higher tear strength; although, this was not the case for all films. The increasing dwell time of the dipping process does not significantly affect tensile or tear strength. However, after post-processing, the mechanical characteristics of **LC** films showed better tensile and tear strengths than those without leaching. As a result, double-layer **LC** beneath NRL films provides good structural stability for a fluorescence-based glove puncture detection system. Due to the high mechanical qualities and a high emission fluorescence intensity when ripped, **LC3** and **LC7** films offer optimal performance for **LC** glove production. This verification work has shown the potential benefit of **LC** as a wearable glove for in situ torn glove detection. In the future, it is hoped that the **LC** glove can be verified in a clinical study to demonstrate surgical glove tear detection in vivo in a suitable operating environment with appropriate lighting conditions available. Moreover, surgeons require maximum feeling and dexterity to perform the surgery successfully, and the gloves used should significantly prevent infection risk but not at the expense of surgical performance. Our novel integrated glove tearing sensor improves the mechanical strength of the NRL glove using compounds that are generally thought to be safe. Furthermore, it does so in a manner that allows tears to be readily detected in situ by the surgical team, whilst only requiring a single glove to be worn, which should enable the surgeon to perform their duties without compromising tactile sensitivity and manual dexterity.

## Figures and Tables

**Figure 1 polymers-14-03048-f001:**
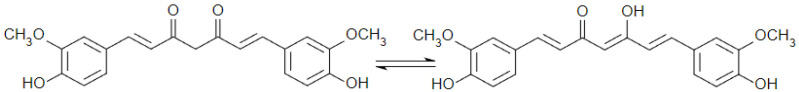
The keto-enol tautomeric structure of curcumin.

**Figure 2 polymers-14-03048-f002:**
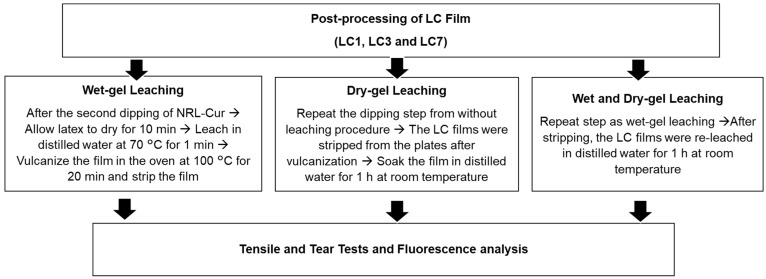
Flow chart of optimised **LC** films post-processing process.

**Figure 3 polymers-14-03048-f003:**
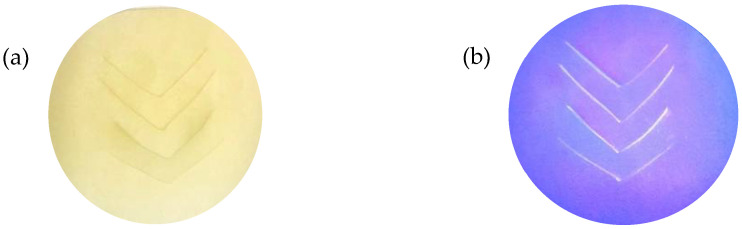
Photograph of **LC** films for solid sample fluorescence analysis (**a**) film under visible light, (**b**) film under UV light.

**Figure 4 polymers-14-03048-f004:**
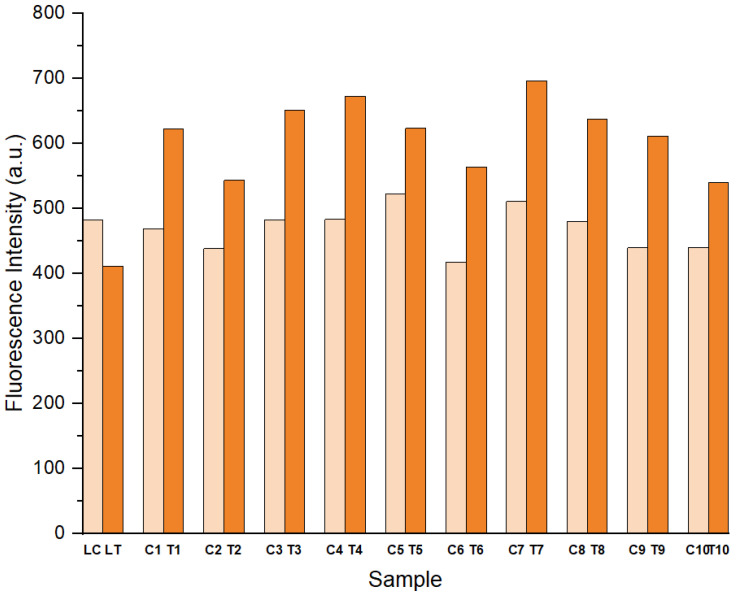
The maximum fluorescence intensity of each sample at emission wavelength 489 nm (**LC** = latex (without torn), **LT** = torn latex, **C** = control (i.e., without any tear), **T** = torn on the film.

**Figure 5 polymers-14-03048-f005:**
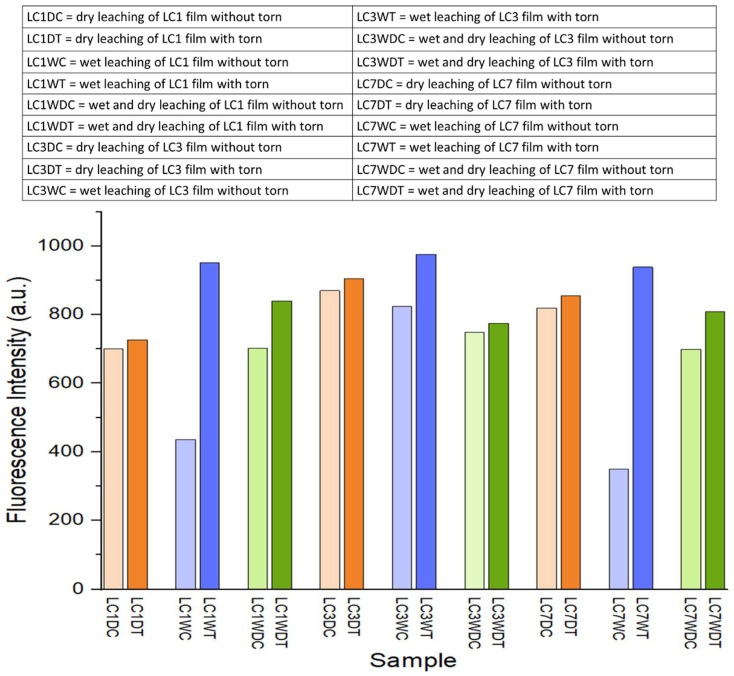
The fluorescence intensity of each sample after leaching at emission wavelength 510 nm.

**Figure 6 polymers-14-03048-f006:**
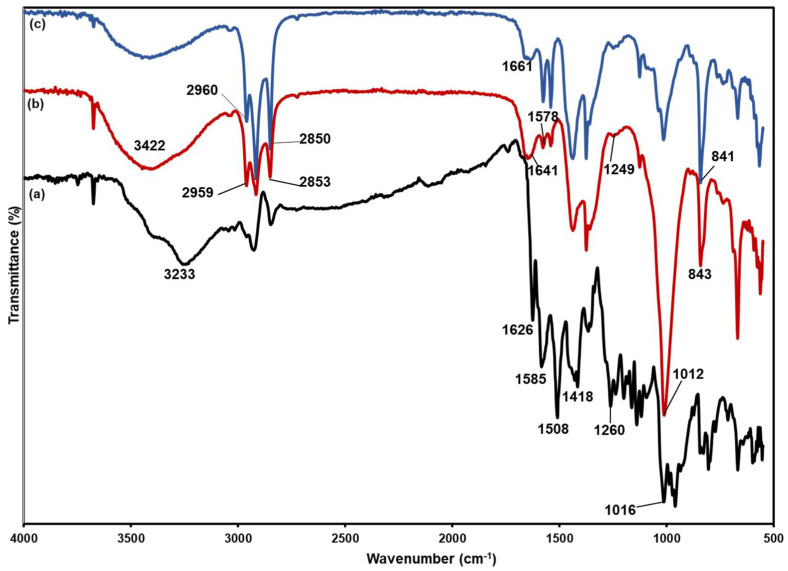
FTIR spectrum of (**a**) curcumin, (**b**) **LC** and (**c**) NRL.

**Figure 7 polymers-14-03048-f007:**
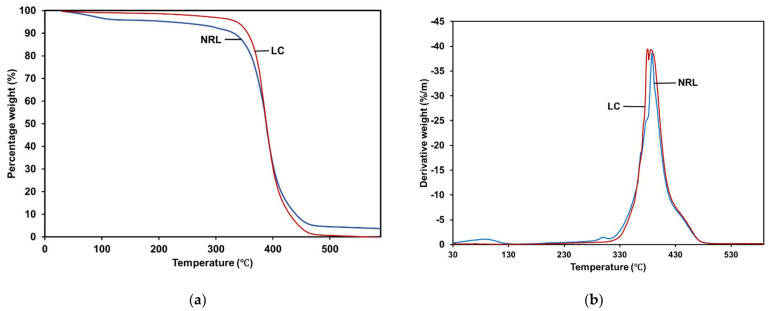
(**a**) TGA and (**b**) derivative thermogravimetry (DTG) curves of NRL and **LC**.

**Table 1 polymers-14-03048-t001:** Dwell time of **LC** film samples and their thickness.

Samples	Dwell Time		Thickness (mm)
	NRL (44%)	NRL-Cur (26%)	MEAN (±SD)
**LC1**	10_10	5_5	0.22 (2.78 × 10^−17^)
**LC2**	10_10	5	0.21 (2.78 × 10^−17^)
**LC3**	10_5_10	5_5	0.24 (4.90 × 10^−3^)
**LC4**	10_5_10	5	0.23 (4.00 × 10^−3^)
**LC5**	15_15	5_5	0.24 (8.94 × 10^−3^)
**LC6**	15_15	5	0.23 (4.90 × 10^−3^)
**LC7**	15_5_15	5_5	0.26 (4.90 × 10^3^)
**LC8**	15_5_15	5	0.25 (4.30 × 10^3^)
**LC9**	15_15	10_5_10	0.26 (0.00)
**LC10**	15_15	10_10	0.25 (4.71 × 10^−3^)
**L**	10_10		0.15 (0.00)

**L** = NRL film, **LC** = double coating of NRL with NRL-Cur film, Dwell time = 10_10 means 10 s dip in, and 10 s take out, while 10_5_10 means 10 s dip in, 5 s hold and 10 s take out, (±SD) = Standard Deviation.

**Table 2 polymers-14-03048-t002:** Mechanical properties of **LC** films before leaching.

Samples	Tensile Strength (MPa)	Tear Strength (N/mm)
	Mean (±SD)	Mean (±SD)
**LC1**	19.10 (0.96)	40.55 (10.15)
**LC2**	18.09 (0.69)	45.78 (9.49)
**LC3**	18.82 (1.76)	40.13 (7.87)
**LC4**	18.96 (1.76)	41.25 (8.60)
**LC5**	19.14 (0.73)	41.72 (1.66)
**LC6**	19.13 (1.47)	44.23 (9.51)
**LC7**	18.67 (0.81)	39.84 (4.23)
**LC8**	18.31 (0.77)	43.05 (14.41)
**LC9**	20.10 (0.45)	39.97 (13.08)
**LC10**	18.35 (0.55)	40.20 (3.19)
**L**	17.92 (0.17)	37.73 (6.14)

**L** = NRL film, **LC** = double coating of NRL with NRL-Cur film, (±SD) = Standard Deviation.

**Table 3 polymers-14-03048-t003:** Mechanical properties of **LC1, LC3** and **LC7** film post-processing (wet-gel leaching, dry-gel leaching, and wet and dry-gel leaching).

Leaching Conditions	LC1		LC3		LC7	
	Tensile (MPa)	Tear (N/mm)	Tensile (MPa)	Tear (N/mm)	Tensile (MPa)	Tear (N/mm)
Unleached	19.10	40.55	18.82	41.72	18.67	39.84
Wet	21.23	52.02	23.36	54.15	21.98	57.91
Dry	21.60	53.86	23.04	54.30	21.09	55.49
Wet + Dry	21.90	58.60	23.81	65.47	22.74	63.85

**L** = NRL film, **LC** = double coating of NRL with NRL-Cur film.

**Table 4 polymers-14-03048-t004:** Images of **LC1, LC3,** and **LC7** torn film before and after leaching.

LC Films	Without Leaching	Wet Leaching	Dry Leaching	Wet-Dry Leaching
**LC1**	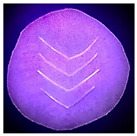	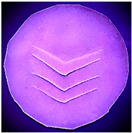	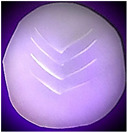	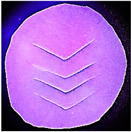
**LC3**	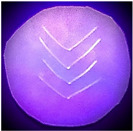	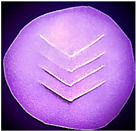	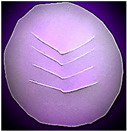	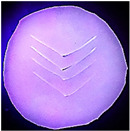
**LC7**	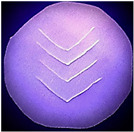	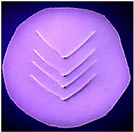	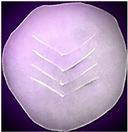	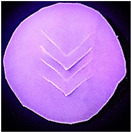

**Table 5 polymers-14-03048-t005:** The thermal stability comparison of NRL and **LC**.

Type of Film	T_5%_ (°C)	T_50%_ (°C)	T_75%_ (°C)	Total Weight Loss (%)
NRL	211.92	387.92	405.92	96.329
**LC**	336.92	388.92	403.92	100

## Data Availability

All the actual data are presented in the manuscript.

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
