# Peer review of "Wearable Natural Rubber Latex Gloves with Curcumin for Torn Glove Detection in Clinical Settings"

_polymers, 2022, doi:10.3390/polym14153048_

Round 1
Reviewer 1 Report
Dear Authors,
In this paper authors have prepared Curcumin based latex glove where torn or punctured can be detected by fluorescence measurements. Authors have also studied mechanical and thermal properties of the latex films to fulfill the requirements for practical applications. Although this paper contains some good results and have well discussion, however, I have few suggestions to include as bellow before publication
1. What is the advantage of fluorescence based text compared to normal air pressure test for detection of torn gloves?
2. In Page 6, line 244 “thickness (25-26 mm)” should be corrected.
3. Why the char residue of LC film is zero but the char residue of NRL film is about 3-4% in TGA? This should be explained.
Author Response
Thank reviewer for your valuable comments on our paper. "Please see the attachment".

Reviewer 2 Report
Comments and Suggestions for Authors
The manuscript by Sarih et al. reports wearable natural rubber latex gloves with curcumin for torn glove detection. The novel and low-cost detection using fluorescence from curcumin in natural rubber latex is proposed. This approach indicates an effective method to detect glove tear or perforation. The contents are informable and logical, but the referee suggests minor revision before it can be published. There are some points that concern the reviewer as follows:
L. 71 [3]: Is this reference correct?
L. 103 6000 pm: 6000 rpm?
L. 123 thickness gauge: More detailed information is needed, such as manufacturer, model name, and type.
L. 147 2.4 Post-processing of LC film: This subsection seems a little bit complicated. Is it possible to add some drawing or flow chart?
L. 169 tensile tester machine (Instron machine): More detailed information is needed, such as model name, and type.
L. 177 The excitation wavelength was excited at 380 nm: => "The excitation wavelength was 380 nm" or "The excitation wavelength was set at 380 nm."
L. 178 the fluorescence emission spectrum was scanned from 400 to 600 nm.: The scanning time is needed. The breeching effect was negligible? What part of the sample did the authors monitor by the spectrometer? The whole of each sample?
L. 190 using an Alpha FTIR Spectrometer (Bruker Optics, MA, USA): => using a FTIR Spectrometer (Bruker Optics, MA, USA, Model Alpha)
L. 191 4,000 to 500 cm-1: It is better to add wavelength information like "(2.5 to 20 um)."
L. 194 a Perklin-Elmer Pyris 6 TGA analyser (Perkin Elmer, MA, USA): => a TGA analyser (Perkin Elmer, MA, USA, Model Pyris 6)
L. 236 Moreover, when compared based on the dwell time of NRL latex dipping, most of the LC films do not show much difference, especially with the similar dwell time of LC mixture dipping.: Is it possible to show the difference using standard deviation/standard error?
L. 241 Based on this observation, the dwell time of NRL dipping that was held for 5 secs during dipping produced a better fluorescence emission appearance of torn on film than other dwell times.: The authors should show why the "5s" samples emitted stronger fluorescence.
L. 244 the thickness (25-26 mm): => the thickness (0.25-0.26 mm) Is it right?
L. 246 [38]: Is this reference correct? Is it enough to show ASTM D3577-78a?
L. 253 the optimum films (LC1, LC3 and LC7): The authors should explain why these samples are optimum.
L. 258 [39]: The referee could not reach this reference.
Table 3: The used sample number should be mentioned.
L. 270 510 nm: The authors should explain why the wavelength was changed from 489 nm in Fig. 3 to 510nm -watching the Figs. S1-S10, the referee feels it is better to use ~490 nm with some bandwidth because the fluorescence spectra were not stable, where the peak wavelengths were shifting.
Table 4: Were all violet colors of sample images caused by the excitation wavelength of 380 nm?
Figure 5: The authors should mark the mentioned lines in the manuscript in the spectra. It is better to add the wavelengths to the wavenumber axis for the reader's understanding.
L. 314 [44, 45]: The reference [44] is suitable?
L. 318 at 358℃: => at 538℃?
Refrences
Ref [39] : The referee could not reach this reference as mentioned in L. 258.
The referee's other questions:
1. Why do the fluorescence intensities increase with the tear?
2. Why do the fluorescence intensities increase by the leaching procedure?
3. The authors indicate various data but did not supply the organic linkage of these data in the manuscript.
4. How can this technique detect glove tear or perforation in an operating theatre? The authors should indicate some ideas.
Author Response
Thank reviewer for your valuable comments on our paper. "Please see the attachment."
